The fungal pathogen Batrachochytrium salamandrivorans is not detected in wild and captive amphibians from Mexico

Basanta M. Delia 1 2 3
Avila-Akerberg Victor 4
Byrne Allison Q. 5
Castellanos-Morales Gabriela 6
González Martínez Tanya M. 2
Maldonado-López Yurixhi 7
Rosenblum Erica Bree 5
Suazo-Ortuño Ireri 8
Parra Olea Gabriela gparra@ib.unam.mx 9
Rebollar Eria A. ea.rebollar@gmail.com rebollar@ccg.unam.mx 1
1 Centro de Ciencias Genómicas, Universidad Nacional Autónoma de México , Cuernavaca , Morelos , Mexico
2 Facultad de Ciencias, Universidad Nacional Autónoma de México , Coyoacán , Ciudad de México , Mexico
3 Department of Biology, University of Nevada Reno , Reno , Nevada , United States of America
4 Instituto de Ciencias Agropecuarias y Rurales, Universidad Autónoma del Estado de México , Toluca , Estado de México , Mexico
5 Department of Environmental Science, Policy, and Management, University of California , Berkeley , CA , United States of America
6 Departamento de Conservación de la Biodiversidad, Colegio de la Frontera Sur Unidad , Villahermosa , Tabasco , México
7 CONACYT-Instituto de Investigaciones sobre los Recursos Naturales, Universidad Michoacana de San Nicolás de Hidalgo , Morelia , Michoacán , Mexico
8 Instituto de Investigaciones sobre los Recursos Naturales, Universidad Michoacana de San Nicolás de Hidalgo , Morelia , Michoacán , Mexico
9 Instituto de Biología, Universidad Nacional Autónoma de México , Ciudad de México , Ciudad de México , Mexico
Mora-Montes Héctor
Electronic publication date: 2022 Oct 3
Publication date: 2022
Volume: 10
Electronic Location ID: e14117
Received 2022 Jul 13; Accepted 2022 Sep 4
Copyright: ©2022 Basanta et al.
Copyright year: 2022
Copyright holder: Basanta et al.
License: This is an open access article distributed under the terms of the Creative Commons Attribution License, which permits unrestricted use, distribution, reproduction and adaptation in any medium and for any purpose provided that it is properly attributed. For attribution, the original author(s), title, publication source (PeerJ) and either DOI or URL of the article must be cited.
License URL: https://creativecommons.org/licenses/by/4.0/

Keywords: Chytridiomycosis, Disease ecology, Amphibians, B. salamandrivorans

Funding: CONACyT Project CF-2019/373914 UC MEXUS-CONACyT CN 18-127 Programa de Apoyo a Proyectos de Investigación e Innovación Tecnológica (PAPIIT-UNAM) IN205521 Centro de Ciencias Genómicas, Universidad Nacional Autónoma de México (UNAM) CONACyT in the form of a postdoctoral fellowship Project CF-2019/373914 This research was supported by grants from CONACyT Project CF-2019/373914 granted to Eria A. Rrebollar, the UC MEXUS-CONACyT CN 18-127 granted to Gabriela Parra Olea and Erica B. Rosenblum, and Programa de Apoyo a Proyectos de Investigación e Innovación Tecnológica (PAPIIT-UNAM) IN205521 granted to Gabriela Parra Olea. There was no additional external funding received for this study. Maria Delia Basanta received funding from the Centro de Ciencias Genómicas, Universidad Nacional Autónoma de México (UNAM), and CONACyT in the form of a postdoctoral fellowship (Project CF-2019/373914). The funders had no role in study design, data collection and analysis, decision to publish, or preparation of the manuscript.

==============================
The recent emergence of the pathogen Batrachochytrium salamandrivorans (Bsal) is associated with rapid population declines of salamanders in Europe and its arrival to new areas could cause dramatic negative effects on other amphibian populations and species. Amphibian species, present in areas with high amphibian diversity such as Mexico, could be highly threatened due to the arrival of Bsal, particularly salamander species which are more vulnerable to chytridiomycosis caused by this pathogen. Thus, immediate surveillance is needed as a strategy to efficiently contend with this emerging infectious disease. In this study, we analyzed 490 wild and captive amphibians from 48 species across 76 sites in the North, Central, and South of Mexico to evaluate the presence of Bsal. Amphibians were sampled in sites with variable degrees of amphibian richness and suitability for Bsal according to previous studies. From the 76 sampling sites, 10 of them were located in areas with high amphibian richness and potential moderate to high Bsal habitat suitability. We did not detect Bsal in any of the samples, and no signs of the disease were observed in any individual at the time of sampling. Our results suggest that Bsal has not yet arrived at the sampled sites or could be at low prevalence within populations with low occurrence probability. This is the first study that evaluates the presence of Bsal in different regions and amphibian species in Mexico, which is the second most diverse country in salamander species in the world. We highlight the risk and the importance of continuing surveillance of Bsal in Mexico and discuss control strategies to avoid the introduction and spread of Bsal in the country.

Introduction

Chytridiomycosis is an emerging disease caused by the fungal pathogens Batrachochytrium dendrobatidis (Bd) and B. salamandrivorans (Bsal) and is considered one of the principal causes of amphibian population declines worldwide (Longcore, Pessier & Nichols, 1999; Skerratt et al., 2007; Martel et al., 2013; Scheele et al., 2019; Fisher & Garner, 2020). Whereas Bd has been associated with worldwide population declines since the 1970s (Skerratt et al., 2007; Cheng et al., 2011; Fisher & Garner, 2020), the recent emergence of Bsal is associated with rapid population declines of the European salamander Salamandra salamandra (Martel et al., 2014; Stegen et al., 2017). Currently, the known distribution of Bsal is restricted to several countries in Asia and Europe (Martel et al., 2014; Laking et al., 2017; Yuan et al., 2018; Lötters et al., 2020). Amphibians from Asia have shown resistance and tolerance to Bsal infections, whereas European salamanders have suffered population declines and have shown severe symptoms due to Bsal infections (Martel et al., 2013; Martel et al., 2014). The tolerance or resistance to Bsal infections in Asiatic amphibians may be due to a long co-evolution between the pathogen and their hosts, whereas a recent introduction of this pathogen in naïve salamander populations could be the reason for the declines detected in Europe (Martel et al., 2014; Laking et al., 2017).

The pet trade has been suggested as one of the primary causes for the spread of Bsal to naïve areas, and experimental studies have demonstrated that Bsal infections are lethal to several North American salamanders such as Notophthalmus viridescens, Chiropterotriton spp. and Aquiloeurycea cephalica (Martel et al., 2014; North American Bsal Task Force, 2022). The pathogen Bsal has been detected in imported amphibians from Asia to Europe that did not show signs of disease (Cunningham et al., 2015; Gray et al., 2015; Nguyen et al., 2017; Yuan et al., 2018). Anurans have been considered more tolerant to Bsal infections than salamanders, acting as reservoirs and carrier species of the pathogen (Martel et al., 2014; Stegen et al., 2017; North American Bsal Task Force, 2022). However, recent evidence has shown that anurans can also be susceptible to Bsal infections under an experimental setting (Towe et al., 2021). Additionally, experimental co-infections of Bd and Bsal have shown higher mortalities than infections caused by only one of the pathogens (Longo, Fleischer & Lips, 2019). Considering the potential threat that Bsal represents for amphibian species in naïve regions, there is a considerable concern that Bsal will arrive in new areas such as North America, causing devastating impacts on amphibian diversity (Grant et al., 2015; Gray et al., 2015; North American Bsal Task Force, 2022).

Mexico ranks second in the world in the number of salamander species per country, and seventh in the number of amphibian species in general (AmphibiaWeb, 2022). Moreover, according to the International Union for Conservation of Nature (IUCN) Red List of Threatened Species, 62% of amphibian species in this country are in some category of risk (IUCN, 2022). Additionally, experimental studies of Bsal infections showed lethal effects (high susceptibility) in Mexican plethodontid salamanders (North American Bsal Task Force, 2022), thus more than 133 species of this group distributed in Mexico (AmphibiaWeb, 2022), could likely be susceptible to Bsal infection. It is highly likely that the potential introduction of Bsal to Mexico would represent a serious threat to local and global amphibian diversity. Previous studies have predicted that regions of Mexico such as the Trans Mexican Volcanic Belt, Sierra Madre del Sur, Sierra Madre Oriental, and Northern Oaxaca are places that could become highly suitable for Bsal (Basanta, Rebollar & Parra-Olea, 2019; García-Rodríguez et al., 2022). These studies also have found that the potential suitable habitat areas for Bsal corresponded to areas of high salamander diversity, increasing the risk of Bsal-vulnerable species losses in case the pathogen arrives (Basanta, Rebollar & Parra-Olea, 2019; García-Rodríguez et al., 2022). Thus, Bsal surveillance directed to areas of high suitability for Bsal and high amphibian diversity could be a good strategy to take immediate conservation actions. Recent efforts to monitor the presence of Bsal in the USA and Northern Mexico have not detected the pathogen (Bales et al., 2015; Klocke et al., 2017; Parrott et al., 2017; Newman et al., 2019; Hardman et al., 2020; Waddle et al., 2020; Hill et al., 2021). However, surveillance of Bsal presence in amphibians across Mexico is still lacking.

In this study, our main goal was to survey wild and captive amphibian populations in Mexico to search for the presence of Bsal. We sampled amphibians from different sites across the country with variable degrees of amphibian diversity and Bsal habitat suitability areas based on previous studies (Basanta, Rebollar & Parra-Olea, 2019; García-Rodríguez et al., 2022). This information could be informative for future surveillance efforts in areas of potential risk and identify new areas where pathogen arrival may be more likely to occur in Mexico.

Materials & Methods

Sampling design

We sampled wild amphibian species from North, Central and South Mexico. Additionally, we sampled individuals of Ambystoma mexicanum from a captivity center in Mexico City that includes different populations derived from pet owners and captive reproduction programs. All surveys took place between the years of 2015 to 2021. All wild and captive amphibians were captured using a new inverted plastic bag or sterile plastic container and manipulated them with a new pair of nitrile gloves to avoid cross-contamination (Phillott et al., 2010). Swab sampling was performed using standardized methods (Hyatt et al., 2007) and sterile swabs (MW113 rayon swabs, Medical Wire and Equipment, Corsham, UK). Collection permits were provided by the Secretaría del Medio Ambiente y Recursos Naturales (SEMARNAT) SGPA/DGVS/00947/16, SGPA/DGVS/03038/17, SGPA/DGVS/003513/18, SGPA/DGVS/002176/18, SPA-ENS/305/18, SEDUMA/SP/2738/2018, RBMM. DIREC/208/18, UNAM JJBIB/54/2017, SGPA/DGVS/5673/19, and SGPA/DGVS/02770/21.

Molecular methods

Chytrid DNA was extracted from swabs using two extraction methods that are widely used for chytrid detection: Prepman Ultra and the Qiagen DNeasy Blood and Tissue Kit. For the Prepman extractions, we followed the protocol of Boyle et al. (2004) for 43 samples. Extractions from Prepman were diluted to a concentration of 1:10 to avoid any inhibition during the qPCR process (Boyle et al., 2004). Given that Qiagen extractions result in a higher quantity and quality of extracted DNA (Cheng et al., 2011), we switched for this approach for 447 samples and followed the protocol of the manufacturer with an additional lysozyme step (Rebollar et al., 2016).

We used the extracted DNA from swabs to detect Bsal through two different methods: quantitative TaqMan polymerase chain reaction (qPCR) as described in Martel et al. (2013), and an amplicon sequencing approach using the Fluidigm Juno system as described in Byrne et al. (2017) (Table S1). From 490 samples analyzed, 384 were tested with qPCR and 106 were sequenced via Fluidigm. For the qPCR, each sample was assayed in duplicate with one negative control (sterile water), one positive control (DNA extraction of Bsal isolate donated by Dr. Vance T. Vredenburg, San Francisco State University), and four standards of DNA made with a synthetic fragment of the 5.8S-ITS1 region of Bsal (Martel et al., 2013): 1,100,1000, and 10,000 ITS Bsal equivalent copies. We considered a positive detection of Bsal DNA if a detectable signal existed at 40 or fewer qPCR cycles. For the amplicon sequencing approach, samples were sequenced as described in Basanta et al. (2021a). Briefly, DNA extracts were first cleaned using an isopropanol precipitation and preamplified using two pools of 96 primer pairs. Samples were then loaded into a Fluidigm Juno LP 192.24 IFC (Fluidigm, Inc., South San Francisco, CA, USA) which performed microfluidic PCR amplification of 192 amplicons, one of which was designed to amplify the ITS1 region of Bsal. After amplification, samples were barcoded and pooled for sequencing on an Illumina MiSeq Lane using the Micro 300 bp paired-end kit.

Salamander richness and Bsal suitability models

We estimated the amphibian richness for each site sampled. For this, we performed a richness map at a resolution of 30 arcseconds (∼1 km2) using the distribution range maps of amphibians from Mexico (IUCN, 2022) and the fasterize package (Ross, Sumner & Alliance, 2020) in R v.3.6.1 (R Core Team, 2019). Then, we extracted the richness values for each site using QGIS v3.8.3 (http://www.qgis.org). Amphibian richness can be used as a predictor of the diversity in the sampled sites and could inform how many species could be affected by Bsal. We also revised the risk category according to the IUCN (IUCN, 2022) for all amphibian species analyzed. Additionally, we estimated the potential risk of Bsal on each sampled site by extracting the Bsal suitability values from the model obtained in García-Rodríguez et al. (2022) using QGIS v3.8.3 (http://www.qgis.org). Because the logistic output of the suitability map of Bsal ranges from 0 to 1, with 0 indicating unsuitable habitat and 1 indicating the highest suitability, we also categorized them into four categories of suitability (not suitable, low suitability, moderate suitability, high suitability) according to Basanta, Rebollar & Parra-Olea (2019).

Results

We collected 490 samples of 48 species from nine families across 76 sites in Mexico (Fig. 1, Table 1, Table S1). Of the total number of samples, 463 were from wild individuals and 27 from captive individual, none were Bsal-positive by our laboratory methods (Table S1). No signs of the disease were observed in any individual at the time of sampling.

Figure 1 Sampled localities in Mexico analyzed for Batrachochytrium salamandrivorans (Bsal) detection.

(A) Map of amphibian richness in Mexico constructed using amphibian distribution maps of the IUCN (2022). (B) Map of Bsal habitat suitability obtained from García-Rodríguez et al. (2022) and categorized following Basanta, Rebollar & Parra-Olea (2019). Circles represent sampled sites for wild amphibians, and the white square represents the sample site for captive amphibians.

Of the 48 species sampled, 24 of them were endemic to Mexico, and according to the risk categories by the IUCN (IUCN, 2022), 23 are Critically Endangered (CR), Endangered (EN), Vulnerable (VU), or Near Threatened (NT) (Table S1). Our sampled sites in the wild were located in areas where 163 amphibian species are distributed (39% of the total amphibian diversity in Mexico). Of the 76 sampled sites in the wild, 50 were from areas of high species richness, hosting between 13 and 28 amphibian species (Fig. 1, Table S1). The rest of the sites (n = 26) were from areas of low-medium richness hosting between seven to 12 amphibian species (Table S1). Additionally, six species (84 of 490 [17%] individuals examined) were sampled from 10 sites with moderate to high Bsal habitat suitability. Finally, four sites had high species richness and moderate to high Bsal habitat suitability. The species surveyed in these four sites were Ambystoma granulosum (EN), A. rivulare (DD), Chiropterotriton totonacus (CR) and Rana montezumae (LC), which are distributed in the Trans-Mexican Volcanic Belt region and are all endemic to Mexico (Table S1).

Table 1 Sample size and taxonomic identification of amphibian species analyzed in this study to detect Batrachochytrium salamandrivorans in Mexico.

Species	N	Order	Family	Predicted Bsal vulnerability	Bd infection known in Mexico	References of Bd detection in Mexico	
Anaxyrus boreas	3	Anura	Bufonidae	Tolerant	Positive	Peralta-García et al. (2018) and Basanta et al. (2021a)	
Incilius sp.	1	Anura	Bufonidae	Tolerant	Positive	Basanta et al. (2021a)	
Craugastor matudai	2	Anura	Craugastoridae	No data	Positive	Basanta et al. (2021a)	
Craugastor pygmaeus	1	Anura	Craugastoridae	No data	Positive	Familiar-López (2010) and Basanta et al. (2021a)	
Craugastor rhodopis	2	Anura	Craugastoridae	No data	Positive	Murrieta-Galindo et al. (2014) and Basanta et al. (2021a)	
Craugastor sp.	3	Anura	Craugastoridae	No data	Positive	Basanta et al. (2021a)	
Eleutherodactylus sp.	1	Anura	Eleutherodactylidae	No data	Positive	Basanta et al. (2021a)	
Dryophytes arenicolor	2	Anura	Hylidae	Tolerant	Positive	Basanta et al. (2021a)	
Hyliola cadaverina	3	Anura	Hylidae	Tolerant	Positive	Peralta-García et al. (2018) and Basanta et al. (2021a)	
Hyliola regilla	22	Anura	Hylidae	Tolerant	Positive	Luja et al. (2012), Peralta-García et al. (2018) and Basanta et al. (2021a)	
Plectrohyla matudai	1	Anura	Hylidae	Tolerant	Positive	Muñoz Alonso (2010) and Basanta et al. (2021a)	
Plectrohyla sp.	1	Anura	Hylidae	Tolerant	Positive	Basanta et al. (2021a)	
Ptychochyla zophodes	1	Anura	Hylidae	Tolerant	Positive	Basanta et al. (2021a)	
Ptycohyla sp.	2	Anura	Hylidae	Tolerant	Positive	Basanta et al. (2021a)	
Scinax staufferi	5	Anura	Hylidae	Tolerant	Positive	Basanta et al. (2021a)	
Smilisca baudinii	2	Anura	Hylidae	Tolerant	Positive	Familiar-López (2010), Muñoz Alonso (2010) and Basanta et al. (2021a)	
Tlalocohyla loquax	1	Anura	Hylidae	Tolerant	Positive	Muñoz Alonso (2010)	
Tripion spinosus	1	Anura	Hylidae	Tolerant	Positive	Basanta et al. (2021a)	
Leptodactylus melanonotus	3	Anura	Leptodactylidae	No data	Positive	Cortés-García (2014), Basanta et al. (2021a) and Basanta et al. (2021b)	
Agalychnis callydrias	3	Anura	Phyllomedusidae	No data	Positive	Basanta et al. (2021a)	
Agalychnis dacnicolor	4	Anura	Phyllomedusidae	No data	Positive	García-Feria et al. (2017), Basanta et al. (2021a) and Basanta et al. (2021b)	
Agalychnis moreletii	5	Anura	Phyllomedusidae	No data	Positive	Frías-Alvarez et al. (2008), Basanta et al. (2021a) and Basanta et al. (2021b)	
Rana berlandieri	3	Anura	Ranidae	Tolerant	Positive	Murrieta-Galindo et al. (2014), Feria, Brousset & Olivares (2019), Hernández-Martínez et al. (2019) and Basanta et al. (2021a)	
Rana catesbeiana	1	Anura	Ranidae	Tolerant	Positive	Hernández-Martínez et al. (2019) and Basanta et al. (2021a)	
Rana draytonii	1	Anura	Ranidae	Tolerant	Positive	Peralta-García et al. (2018) and Basanta et al. (2021a)	
Rana montezumae	12	Anura	Ranidae	Tolerant	Positive	Frías-Alvarez et al. (2008), García-Feria et al. (2017) and Basanta et al. (2021a)	
Rana neovolcanica	6	Anura	Ranidae	Tolerant	Positive	Frías-Alvarez et al. (2008) and Basanta et al. (2021a)	
Rana sierramadrensis	2	Anura	Ranidae	Tolerant	Positive	Familiar-López (2010) and Basanta et al. (2021a)	
Rana sp.	7	Anura	Ranidae	Tolerant	Positive	Basanta et al. (2021a)	
Ambystoma altamirani	78	Caudata	Ambystomatidae	Tolerant	Positive	Frías-Alvarez et al. (2008) and Basanta et al. (2021b)	
Ambystoma andersoni	30	Caudata	Ambystomatidae	Tolerant	Positive	Basanta, Rebollar & Parra-Olea (2019)	
Ambystoma dumerilii	16	Caudata	Ambystomatidae	Tolerant	Positive	Basanta, Rebollar & Parra-Olea (2019)	
Ambystoma flavipiperatum	1	Caudata	Ambystomatidae	Tolerant	Positive	Basanta, Rebollar & Parra-Olea (2019)	
Ambystoma granulosum	38	Caudata	Ambystomatidae	Tolerant	Positive	Frías-Alvarez et al. (2008)	
Ambystoma mexicanum	27	Caudata	Ambystomatidae	Tolerant	Positive	García-Feria et al. (2017)	
Ambystoma ordinarium	100	Caudata	Ambystomatidae	Tolerant	Positive	Basanta et al. (unpublished data)	
Ambystoma rivulare	2	Caudata	Ambystomatidae	Tolerant	Positive	Frías-Alvarez et al. (2008) and Basanta, Rebollar & Parra-Olea (2019)	
Ambystoma taylori	24	Caudata	Ambystomatidae	Tolerant	Positive	Basanta et al. (unpublished data)	
Ambystoma velasci	60	Caudata	Ambystomatidae	Tolerant	Positive	Frías-Alvarez et al. (2008) and García-Feria et al. (2017)	
Aquiloeurycea cafetalera	1	Caudata	Plethodontidae	Lethal	Positive	Parra-Olea et al. (unpublished data)	
Bolitoglossa franklini	1	Caudata	Plethodontidae	Lethal	Positive	Basanta et al. (2021a)	
Bolitoglossa occidentalis	3	Caudata	Plethodontidae	Lethal	Positive	Basanta et al. (2021a)	
Bolitoglossa platydactyla	1	Caudata	Plethodontidae	Lethal	Positive	Basanta et al. (2021a)	
Chiropterotriton totonacus	1	Caudata	Plethodontidae	Lethal	No data	No data	
Dendrotriton xolocalcae	2	Caudata	Plethodontidae	Lethal	Positive	Muñoz Alonso (2010)	
Parvimolge townsendi	1	Caudata	Plethodontidae	Lethal	Positive	Parra-Olea et al. (unpublished data)	
Pseudoeurycea leprosa	2	Caudata	Plethodontidae	Lethal	Positive	Van Rooij et al. (2011), Mendoza-Almeralla et al. (2016) and Basanta et al. (2021a)	
Pseudoeurycea longicauda	1	Caudata	Plethodontidae	Lethal	Positive	Van Rooij et al. (2011)	
Notes.

Fifth column indicates the predicted Bsal vulnerability based on infection trials performed in species from the same amphibian family (Martel et al., 2014; North American Bsal Task Force, 2022). Last two columns show data on Bd infection detected in Mexico.

Discussion

This is the first study that evaluates the presence of Bsal in amphibian species across different regions of Mexico. We did not detect Bsal in any of the amphibian skin samples from captive or wild individuals from North, Central and Southern Mexico. The non-detection of Bsal in those areas suggests that to date, Bsal has not yet arrived to these areas. In agreement with our results, previous studies have not detected Bsal in the USA and Northern Mexico (Bales et al., 2015; Klocke et al., 2017; Parrott et al., 2017; Newman et al., 2019; Hardman et al., 2020; Waddle et al., 2020; Hill et al., 2021), which could indicate that Bsal is still absent in North America, or that Bsal is at very low prevalence within these populations and has a low detection probability. Because the risk of this pathogen is high for amphibians and its arrival to America is imminent, we highlight the importance of continuous surveillance of Bsal in areas of potential risk and new areas where the pathogen arrival is more likely to occur.

To date, Bsal detections from animals in the wild remain restricted to Europe and Asia (Martel et al., 2014; Laking et al., 2017; Yuan et al., 2018; Lötters et al., 2020), and amphibians from the pet trade from Asia have been found infected by Bsal (Cunningham et al., 2015; Nguyen et al., 2017; Yuan et al., 2018; Martel et al., 2020). Thus, individuals from trade markets are considered as the principal potential source for the spread of this pathogen to naïve areas (Gray et al., 2015). In our study, we sampled captive individuals of Ambystoma mexicanum. This species is one of the most traded amphibians worldwide (Carpenter et al., 2014), and the potential presence of Bsal in this species could threaten not only amphibians from Mexico but also from other countries that have a high amphibian trade such as the USA and Canada. Future surveillance should include the screening of amphibians that were potentially imported.

Considering the high amphibian richness present in Mexico, the high number of species in some category of risk, and the potential vulnerability in plethodontid species from laboratory infection trials, the introduction of Bsal into the country should be considered a major threat to amphibian biodiversity. Our survey includes regions that have a high amphibian taxonomic and functional diversity (Ochoa-Ochoa et al., 2020; García-Rodríguez et al., 2022), and/or have a high or moderate Bsal habitat suitability: the Sierra Madre Oriental, the extreme eastern Trans-Mexican Volcanic Belt, southeastern Sierra Madre del Sur, mountains of Chiapas in Mexico (García-Rodríguez et al., 2022). One example are the Mexican bolitoglossine salamanders Chiropterotriton spp. and Aquiloeurycea cephalica which are highly vulnerable to Bsal as evidenced by their high mortality in experimental infections (North American Bsal Task Force, 2022). Thus, the arrival of Bsal could cause decreases in populations and could dramatically reduce salamander bolitoglossine biodiversity.

Moreover, we sampled 41 of the 103 amphibian species that have tested positive to Bd infection in Mexico (Table 1; López-Velázquez, 2018; Basanta et al., 2021a). The potential co-infection of Bd and Bsal to these species could affect them severely (Longo, Fleischer & Lips, 2019). We suggest continuing the surveillance and increasing the efforts in the areas and species with high risk (i.e., areas with high Bsal suitability, high amphibian richness, and Bd presence), however, it is also important to start strict Bsal surveillance and monitoring at all entry points for exotic species for trade.

Understanding the potential entry sites and invasion routes of Bsal may help in designing efficient surveillance and conservation strategies. Pathogen transmission could be influenced by the amphibian species-specific Bsal susceptibility to infection. Thus, knowledge of species susceptibility to Bsal and information on its spatial transmission (Kearney & Porter, 2009; Malagon et al., 2020) may help predicting potential spreading routes in the country if it were to be introduced. Previous experimental studies have shown that salamanders infected by Bsal show clinical signs of chytridiomycosis, whereas anurans were asymptomatic and may act as potential reservoirs of the pathogen (Martel et al., 2014; North American Bsal Task Force, 2022). In contrast, a recent study found that the Cuban treefrog Osteopilus septentrionalis is highly susceptible to Bsal disease (Towe et al., 2021), demonstrating that infections of Bsal could also affect anuran species under specific scenarios. Experimental research on host Bsal susceptibility in Mexican anuran and salamander species is urgently needed.

Conclusions

The emergence of infectious diseases such as chytridiomycosis in amphibians poses a major threat to wildlife populations (Skerratt et al., 2007; Martel et al., 2013; Fisher & Garner, 2020). To date, there are no effective treatments for Bsal infections in natural populations. The non-detection of Bsal in our study suggests that to date Bsal has not yet arrived in these regions in Mexico. The major challenges are the design of in situ disease containment and mitigation post Bsal arrival (Canessa et al., 2018; Thomas et al., 2019). Control strategies should focus on preventive measures to reduce the introduction risk such as the implementation of biosecurity measures in amphibian trade, and surveillance in areas of potential introduction and high risk. For example, the United States Fish and Wildlife and the government of Canada restricted salamander imports from 2016 and 2017 respectively, which has likely prevented the arrival of Bsal to these countries (https://www.ecfr.gov/current/title-50/part-16; Government of Canada, 2017; North American Bsal Task Force, 2022). It is imperative for a swift legislative change to prevent pathogen spread and implement a strategic plan to prevent and control Bsal invasion in Mexico (North American Bsal Task Force, 2022). For this, it is urgently necessary to establish a working network between Mexican authorities, research groups, pet shops, and local communities to report any signs of disease in any organism to stop the possible spread of Bsal in Mexico.

Supplemental Information

Table S1 Skin swab samples taken for the detection of Batrachochytrium salamandrivorans (Bsal)

Click here for additional data file.

We thank Vance T Vredenburg for donating DNA from a Bsal isolate, and Andrea Jiménez, Laura Márquez-Valdemar, and Mirna García-Castillo for assisting us with the laboratory work. We also thank CIBAC and José Antonio Ocampo for allowing us to sample their captive amphibians, and Enrique Soto, Carolina González-Pardo, Raquel Hernández-Austria, Omar Becerra, Angel Contreras, Aldo López-Velázquez, Ángela Mendoza-Henao, Sean Rovito, Nelson Cerón De la Luz, Eliuth Vega-Orihuela, Sarahi Toribio-Jiménez, Ángel Montaño, Alfredo Gutiérrez-Morales, Omar Chávez, Rusby Contreras-Díaz, Luis Osorio, Alejandro Calzada, Víctor Jiménez, Omar Hernández-Ordóñez, Angel Soto, and Alberto Cruz-Silva for field assistance.

Additional Information and Declarations

Competing Interests

Author Contributions

Field Study Permissions

Data Availability

Gabriela Parra Olea is an Academic Editor for PeerJ.

M. Delia Basanta conceived and designed the experiments, performed the experiments, analyzed the data, prepared figures and/or tables, authored or reviewed drafts of the article, and approved the final draft.

Victor Avila-Akerberg performed the experiments, authored or reviewed drafts of the article, and approved the final draft.

Allison Q. Byrne performed the experiments, authored or reviewed drafts of the article, and approved the final draft.

Gabriela Castellanos-Morales analyzed the data, authored or reviewed drafts of the article, and approved the final draft.

Tanya M. González Martínez performed the experiments, authored or reviewed drafts of the article, and approved the final draft.

Yurixhi Maldonado-López performed the experiments, authored or reviewed drafts of the article, and approved the final draft.

Erica Bree Rosenblum performed the experiments, authored or reviewed drafts of the article, and approved the final draft.

Ireri Suazo-Ortuño performed the experiments, authored or reviewed drafts of the article, and approved the final draft.

Gabriela Parra Olea conceived and designed the experiments, performed the experiments, analyzed the data, authored or reviewed drafts of the article, and approved the final draft.

Eria A. Rebollar conceived and designed the experiments, performed the experiments, analyzed the data, prepared figures and/or tables, authored or reviewed drafts of the article, and approved the final draft.

The following information was supplied relating to field study approvals (i.e., approving body and any reference numbers):

Collection permits were provided by the Secretaría del Medio Ambiente y Recursos Naturales (SEMARNAT): SGPA/DGVS/00947/16, SGPA/DGVS/03038/17, SGPA/DGVS/003513/18, SGPA/DGVS/002176/18, SPA-ENS/305/18, SEDUMA/SP/2738/2018, RBMM. DIREC/208/18, UNAM JJBIB/54/2017, SGPA/DGVS/5673/19, and SGPA/DGVS/02770/21.

The following information was supplied regarding data availability:

The raw data is available as Supplemental File.

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
