# Peer review of "The fungal pathogen Batrachochytrium salamandrivorans is not detected in wild and captive amphibians from Mexico"

_PeerJ, doi:10.7717/peerj.14117_

## Round 0.1 · original submission · Minor Revisions

Two experts in this field assessed your manuscript and found it suitable for publication in this journal after addressing some minor points. These are basically editing issues.

·

Basic reporting

Language and grammar
The English language is written correctly, the ideas are clear and understandable throughout the manuscript. Only have to change some of the words, which I marked in the Word file.

Literature and background
The information provided in the introduction is very accurate, it covers all the points that must be known to understand the work. I consider that some phrases and words should be improved, which I underlined and mentioned in the word file.
As for the discussion, it correctly explains the results and contrasts what they obtained with what was found in other countries such as the United States. This section includes a lot of literature which is related to the research topic.

Structure, figures, and tables
The structure of the article is good and follows the proper order.
Regarding the tables, the titles and descriptions are understandable and give a short summary of what is going to be observed in each of them. I consider that table S1 should be more mentioned in the text, since this is the one that shows the molecular analyzes where they show us the absence of the fungus in the different amphibians studied.

Experimental design

Research question
The research question is clear and well defined, and is congruent with what is concluded.

Rigorous investigation
The approaches and methodologies are appropriate to answer the initial question. The necessary controls were used so that the results had support and credibility, especially in the molecular analyses.

Methods
The methodologies are explained and cited correctly, the details of each of these are mentioned in the text. It should be noted that in molecular analysis the methodologies used are interesting, since they use different strategies which helps them to have more reliable results.

Validity of the findings

This research is novel, it is the first time that an investigation related to pathogenic fungi in different species of amphibians in Mexico has been carried out. It is totally original research. All the findings obtained explain the absence of this fungus in the different sampling points in the country, however, they do not rule out the possibility that it could reach and colonize these individuals.

Conclusions
The conclusions are completely related to the initial research question. They are written correctly, because they present strategies that help mitigate the arrival of this fungus in Mexico. He has many prospects for the future with this research topic.

Additional comments

In general, the research question and the methodologies are interesting and appropriate, as well as the results obtained. The way in which they present the results is understandable, although it is necessary to highlight a little more the molecular information they obtained. All references are related to the subject of study and cite recent information, which is good to be able to compare their results with those obtained in other countries such as the United States.
Finally, it caught my attention that they used two molecular methods for the detection of the fungus. It is well known that on many occasions molecular tools can give us false positives and false negatives. Using various tools could solve this problem and make the data more reliable.

Reviewer 2 ·

Basic reporting

The paper is extremely well written relative to professional presentation, literature, format, and details presented. I have a few very minor suggestions to improve wording or clarify the content of the main text.

Experimental design

The research is well defined, timely, and well conducted with a rigorous design and implementation. It is novel information in a fast developing topic. The information is meaningful and fills knowledge gaps of interest to the discipline.

Validity of the findings

These are important findings, being the first-ever survey for Bsal across multiple amphibian species in Mexico. The data are solid, the conclusions are well stated.

Additional comments

This is a fabulous study, and I applaud the authors for their insight to conduct the work and getting it out to the literature in a very timely way.

Annotated reviews are not available for download in order to protect the identity of reviewers who chose to remain anonymous.

---

## Round 0.2 · accepted · Accept

All the Reviewers concerns were addressed and the manuscript is now suitable for publication in this journal.